# Waveguide-PAINT offers an open platform for large field-of-view super-resolution imaging

Anna Archetti[1], Evgenii Glushkov [2], Christian Sieben[1], Anton Stroganov[1,2], Aleksandra Radenovic [2] & Suliana Manley [1]

Super-resolution microscopies based on the localization of single molecules have been widely adopted due to their demonstrated performance and their accessibility resulting from open software and simple hardware. The PAINT method for localization microscopy offers improved resolution over photoswitching methods, since it is less prone to sparse sampling of structures and provides higher localization precision. Here, we show that waveguides enable increased throughput and data quality for PAINT, by generating a highly uniform ~100 × 2000 $\mu m^2$ area evanescent field for TIRF illumination. To achieve this, we designed and fabricated waveguides optimized for efficient light coupling and propagation, incorporating a carefully engineered input facet and taper. We also developed a stable, low-cost microscope and 3D-printable waveguide chip holder for easy alignment and imaging. We demonstrate the capabilities of our open platform by using DNA-PAINT to image multiple whole cells or hundreds of origami structures in a single field of view.

[1] Laboratory of Experimental Biophysics, Institutes of Physics and Bioengineering, École Polytechnique Fédérale de Lausanne (EPFL), 1015 Lausanne, Switzerland. [2] Laboratory of Nanoscale Biology, Institute of Bioengineering, École Polytechnique Fédérale de Lausanne (EPFL), 1015 Lausanne, Switzerland. These authors contributed equally: Evgenii Glushkov, Christian Sieben. Correspondence and requests for materials should be addressed to S.M. (email: suliana.manley@epfl.ch)

Super-resolution fluorescence microscopies enable the resolution of structures well below the diffraction limit. Among the most commonly used methods are localization microscopies (photoactivated localization microscopy, PALM[1] or stochastic optical reconstruction microscopy, STORM[2]), which typically rely on stochastic photophysical transitions between fluorescent states to separate single fluorophores in space and time. Once separated, molecules can be localized with high precision and their composite positions rendered to create an image. Fluorophores have become highly optimized in their targeting, photostability, and photoswitching[3–5], so that the localization precision can be well-approximated by an inverse square root dependence on the number of photons[6,7]. This implies that it should be routinely possible to resolve structures down to the nanometric scale. Yet, it is important to remember that in localization microscopy, resolution depends not only on localization precision, but also on the density of localizations[8]. A too-low density of localizations results in an undersampled structure, insufficient to resolve its organization even in the case of nanometric localization precisions[9]. This practical limitation of stochastic photoswitching is circumvented by methods that instead use binding and dissociation of fluorescent probes, such as 'points accumulation in nanoscale topography' (PAINT)[10] and extensions thereof which include complementation between target and imager DNA strands in DNA-PAINT[11,12] and protein-fragment probes in 'integrating exchangeable single-molecule localization' (IRIS)[13]. A major advantage of PAINT is that fluorophores in solution can iteratively sample the structures of interest[14], in a process that is only limited by the patience of the experimentalist[15]. Other advantages include the unlimited multiplexing of Exchange-PAINT for multicolor imaging[11], and the possibility to quantify the number of binding sites at each location using qPAINT[16].

To allow binding and dissociation to occur, PAINT requires a reservoir of fluorescent probes (e.g. labelled DNA oligos) in solution surrounding the sample, which brings its own limitations to the method. First, it requires axial optical sectioning to reject the background signal from fluorophores in solution. This can be mitigated in the case of fluorescence enhancement upon binding as for fluorogenic dyes[10], quenching of unbound probes[17] or Förster resonance energy transfer-(FRET) PAINT[18,19], but at the cost of reduced labeling flexibility, increased sample preparation complexity and a potential reduction in localization precision[20,21]. Optical sectioning can be provided by confocal pinholing[22] or total internal reflection fluorescence (TIRF)[23]. However, confocal pinholing also reduces the number of detected signal photons, while TIRF is typically limited in both size and uniformity of illumination. Note that even sophisticated TIRF setups using scanning of the coherent excitation light to reduce interference patterns[24–27] do not eliminate the spatial dependence of the field resulting from a focused Gaussian beam or the field-of-view (FOV) size limitation. Second, the binding rate is set by the concentration of fluorophores in solution, which must be kept low enough that single molecule fluorescence can still be detected over the background. Thus, PAINT generally requires an integration time per localization more than 10x longer than for stochastic photoswitching[28].

We present a waveguide-based approach for PAINT microscopy, waveguide-PAINT, which helps to alleviate both limitations. The waveguide TIRF approach, compared with other approaches such as refractive beam-shaping elements[29–31], introduces additional flexibility including the freedom to image with a low magnification objective[32] (Fig. 1a) and the generation of an evanescent field with a uniform penetration depth[33,34], as well as built-in reference markings for correlative measurements. Our waveguide is designed with an adiabatic taper for single-mode expansion, and fabricated with a process optimized to enable efficient excitation coupling and propagation, resulting in a large, uniform evanescent field for imaging an area up to ~100 × 2000 $\mu m^2$ (Fig. 1f). This effectively permits the parallelization of PAINT measurements to reduce the amount of time required to collect data on many structures. To make this solution accessible, we also share our designs for a compact, mechanically stable microscope setup built from readily-available commercial components and a customized sample holder. Together, these permit reliable coupling of light into the waveguide, provide a reservoir for PAINT imaging solutions, and allow imaging of multiple waveguides on a single chip. We demonstrate our system by performing DNA-PAINT on cellular and DNA origami structures, and achieve high-quality super-resolution images on multiple mammalian (COS-7) cells and thousands of origami in a single field-of-view.

## Results

**Optimized waveguides for large field TIRF illumination.** In designing waveguides, it is important to select appropriate materials to form the core, where the light propagates, and the cladding, which reflects light at the surface of the core to keep it confined. Waveguide TIRF excitation for fluorescence microscopy has previously been demonstrated with high-index waveguide cores fabricated from either tantalum pentoxide ($Ta_2O_5$)[35,36] or silicon nitride ($Si_3N_4$)[33,37], including being used to perform direct STORM imaging. These materials are compatible with fluorescence bioimaging due to their high chemical stability and their transparency in the visible range[38,39]. When embedded in a glass ($SiO_2$) cladding, they produce high refractive index contrast (HIC)[38] waveguides. HIC waveguides strongly confine the propagating electric field (the waveguide mode), which can reduce propagation losses. However, the increased mode interaction at the core-cladding interface results in enhanced losses due to scattering where surface roughness is present[40,41]. Moreover, HIC waveguides can suffer from high coupling losses, mainly due to back-reflections, mismatches between the electrical field properties of the input source and the waveguide mode, and excitation of modes, which are not confined to the waveguide core (the radiation modes)[42–44]. All of these points should be considered during the fabrication of such waveguides.

To create HIC waveguides for TIRF, we decided to fabricate channel waveguides with a rectangular cross-section using $Si_3N_4$ as the core and $SiO_2$ as the cladding material on standard silicon wafers (diameter = 100 mm, thickness = 525 $\mu m$) (Fig. 1b). We then optimized the layout of the waveguide chip and the fabrication process, focusing on four crucial aspects: core material, input interface, transmission losses, and field distribution. This last point presents a particular challenge in wide, inherently multimode waveguides that are needed for large field-of-view imaging.

The core material, silicon nitride, was chosen to be 150 nm thick for the evanescent field to have an appropriate penetration depth of around 85 nm (Supplementary Fig. 1a). It was deposited using a low-pressure chemical vapor deposition (LPCVD) process, which produces either stoichiometric ($Si_3N_4$) or non-stoichiometric ($SiN_x$) material, depending on the proportions of the gases used (dichlorosilane and ammonia). Our observations show that cores formed from stoichiometric silicon nitride with high internal stress give much higher coupling and transmission efficiency in comparison to non-stoichiometric low-stress silicon nitride, likely due to absorption and scattering on impurities and defects in its structure, which are significantly more numerous in the case of non-stoichiometric $SiN_x$ films.

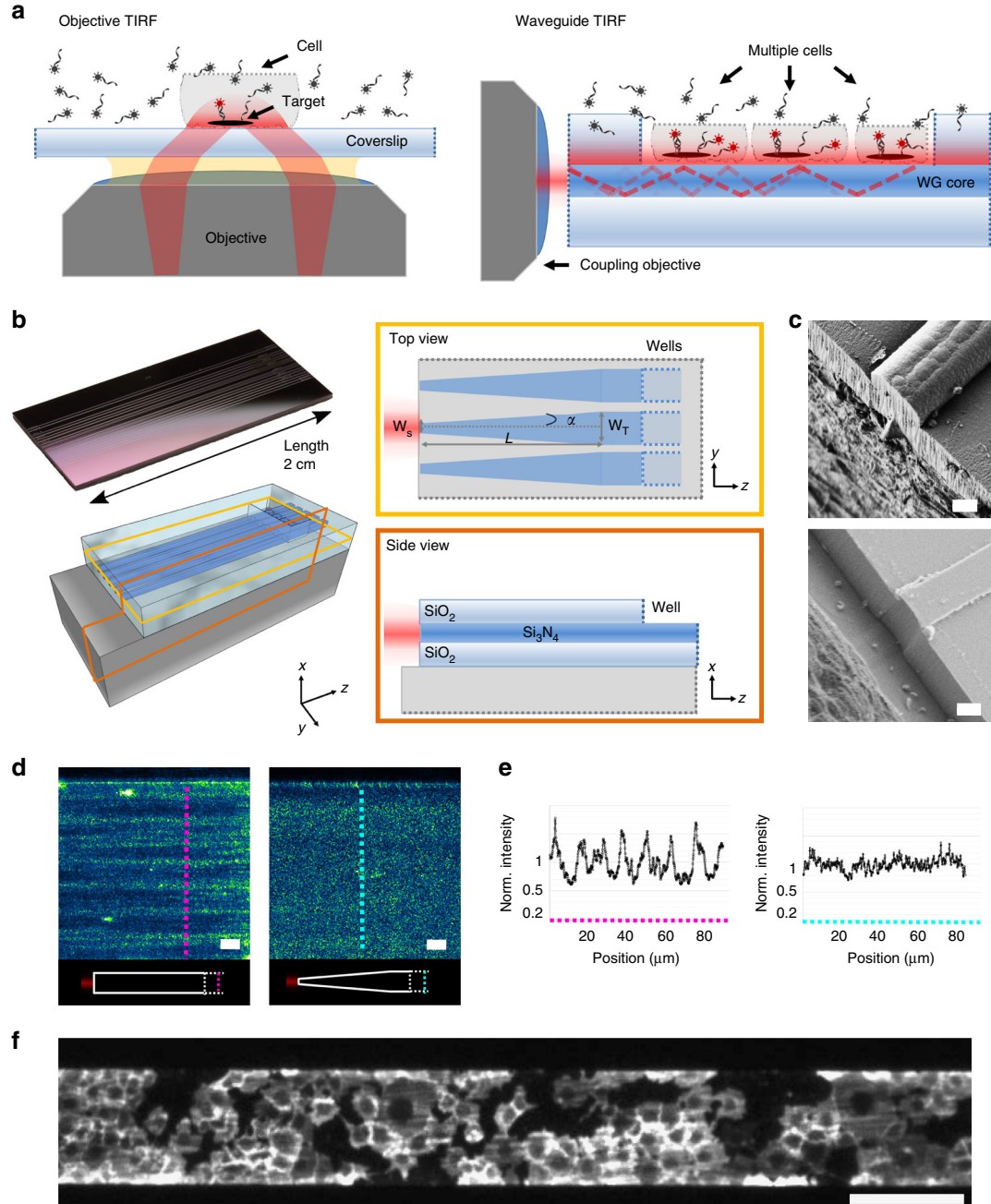

**Fig. 1** Optimized waveguide design enables a uniform and large TIRF illumination. **a** Classical objective TIRF and waveguide TIRF approaches. With objective TIRF, the illumination (red) field size is limited by the objective lens size and magnification and by a roll-off in intensity away from the central axis. In waveguide TIRF, the light (red) undergoes total internal reflection at the interface between the core and the aqueous solution, producing an optical sectioning illumination over the entire waveguide surface (up to 2000 μm in our chips). **b** The chip design includes an inverted nanotaper with a 150 nm input width $w_s$, a 15 mm length $L$, and an expansion rate $\alpha = 0.006$ (yellow box). The waveguide input facet is offset from the substrate etching site (orange box). (See also Supplementary Fig. 2). The waveguide structures appear in reflectance as light-grey stripes on the chip surface (photograph, top left). **c** Scanning electron microscopy of the input facet shows that deep-etching the silicon (Si) substrate after the $Si_3N_4/SiO_2$ layer without the two-step-etching leads to a rough facet (**c** top). Si deep-etching after further lithographic steps—to offset the two etching sites—provides a smooth input facet (**c** bottom). **d** Scattered light from the top waveguide surface in the absence of a taper (left) is less uniform than that with a nanotaper with expansion angle alpha (right). **e** Line profiles (magenta, without taper; cyan, with taper) show modulation depth > 20% and < 12%, respectively. **f** Low magnification (×4) imaging of about 50 COS-7 cells labelled with cholera toxin B conjugated to Alexa 647. Scale bars: 1 μm (**c**), 10 μm (**d**), and 200 μm (**f**)

The second optimization aspect concerned the entrance window to the waveguides where the excitation light was coupled in. Here, we adopted strategies from the field of integrated photonics, where high coupling efficiency and low propagation losses have become central requirements for on-chip waveguides[45,46]. Since losses are strongly affected by surface roughness produced during etching steps, we adopted a two-step lithography and etching process designed to minimize damage to

the input facet[47]. It resulted in a significantly smoother surface, as confirmed by electron micrographs (Fig. 1c, bottom).

To further reduce transmission losses in the waveguide core, we covered its surface with a protective layer of $SiO_2$ (top cladding). The top cladding covers the input and expansion taper, leaving only a selected area exposed by etching, where the sample is placed and imaging is performed (imaging well, Fig. 1b). These two modifications to the fabrication process, which are described in more detail below (see Chip fabrication Methods), led to an increased propagation efficiency (up to 20 times) and reproducibility of coupling light into the on-chip waveguides.

We next optimized the waveguide geometry using numerical simulations (see Numerical simulations and Methods), to achieve both a high coupling efficiency and a large and uniform evanescent field. For efficient coupling, we added a so-called inverted taper coupler (~150 nm wide tip) at the entrance to each waveguide that reduces the mismatch between the properties of the input and the waveguide electrical field (such as the mode size and the effective index; see Supplementary Table 1, Supplementary Fig. 1 and Supplementary Fig. 3). Specifically, by decreasing the waveguide core size, the evanescent field decay length increases, leading to a larger waveguide mode profile that better overlaps with the input beam source profile. A higher overlap between the shape of the input and the waveguide fields leads to a higher coupling efficiency. Moreover, in the case of HIC waveguides, a narrower tip width will produce a waveguide effective index closer to that of the input source, reducing back-reflections originating from mode mismatch.

To achieve a uniform evanescent field in wide multimode waveguides, we adiabatically expanded the fundamental mode from the single-mode waveguide. Using a taper with a slow expansion rate of $\alpha = (w_T - w_s)/2L = 0.006$ (where $L$ is the total taper length, equal to 1.5 cm in our design), similar to[34] (Fig. 1b), we were able to expand from a ~150 nm entrance-tip width ($w_s$) to a 100 μm final width ($w_T$). This expansion rate allows us to restrict the chip length to 2 cm, while the chip width can vary depending on the number of imaging waveguides and the distance between them. Since the footprint of the optimized waveguide chip is only 1 × 2 cm, we were able to reduce the cost per chip by producing multiple chips on a single wafer (up to 24 chips on a 100 mm wafer), each of which can be reused for different experiments (up to ~20 times). The detailed fabrication process for one wafer hosting twelve waveguide chips (each containing twelve imaging waveguides) is described in the Methods section and its layout is provided in Supplementary Fig. 2 and in the data repository (see Data Availability).

The mode propagation after the expansion taper was investigated by simulations and by imaging the light scattered from the top surface of the waveguide. In the absence of a taper, strong fluctuations in intensity are evidence of multimode behavior, whereas with our adiabatic taper, scattering intensity from a rectangular waveguide of identical dimensions appears highly uniform (Fig. 1d, e). We also characterized the uniformity of the TIRF penetration depth by imaging fluorescently-coated beads of a known size[25,48,49]. Assuming a spherical geometry, the distance of points on a bead from the waveguide surface is known, so intensity profiles of individual beads encode the decay length of the evanescent field (Supplementary Fig. 4).

**Realization of a waveguide holder and microscope for PAINT.**
To enable waveguides to be used for PAINT, we needed to build a chip holder that would allow access to the entrance window of the waveguides for coupling with excitation light, but also hold the liquid imaging buffer over the sample to allow fluorophore exchange. Ideally, the holder would also facilitate alignment for easy coupling, and prevent scattered light from entering the detection path. In the previously proposed waveguide platform[33] the chip was secured with a vacuum holder (Waveguide Mount HWV001 Thorlabs) which is more prone to vibration and devoid of the aforementioned features. To fulfill all of these requirements we designed a chip holder (Fig. 2, Supplementary Fig. 5). The holder features a slot tailored to the size of the chip and aligned with the input beam. Chips are easily exchanged, installed and immobilized by means of a removable gate, which is also designed to shield the imaging objective from light scattered at the coupling site.

The PAINT imaging buffer should be held in a reservoir in contact with the sample, which will be prepared on the surface of the waveguide core. We designed a leak-proof chamber by molding and trimming a PDMS strip using the gate itself as a template. This way, the PDMS strip conforms seamlessly to the gate-chip interface. The PDMS was mixed with toner from a printer cartridge, as an affordable way to reduce transmission of light scattered at the entrance window (Supplementary Fig. 6 and Supplementary Fig. 7). We also designed the shape of the reservoir to minimize its volume, while ensuring that the imaging dipping objective could still fully access the sample area. We provide CAD drawings that can be realized either by 3D printing or machining.

Although our waveguides could be imaged on an existing upright microscope, with the simple addition of a coupling objective to introduce excitation light, we designed and built a simple and cost-effective upright microscope with off-the-shelf optomechanical components. The sample holder is supported by a three-axis piezo stage, used for fine adjustment in aligning the entrance window with the coupling objective. In turn, both this stage and the coupling objective are mounted on an $X$-$Y$ platform for imaging different regions of the sample. A mechanically rigid vertical structure suspends the camera and tube lens, while a separate axle with a three-gauge $Z$-adjustment system provides coarse (centimeter range), and fine (micrometric and nanometric) positioning (Fig. 2a and Methods). To better transmit our findings, we share the designs for this microscope as well as a precise workflow pipeline (Supplementary Note 2 and Supplementary Fig. 8), which includes details on key steps such as waveguide excitation coupling and imaging. We found that an effective coupling could be easily established by maximizing the light scattered from the top surface of the waveguide (Supplementary Fig. 5).

**Waveguide DNA-PAINT imaging of cells and DNA origami.**
As light propagates in a waveguide, it is trapped within the high refractive index waveguide core, confined by the lower index cladding. Our design introduces imaging wells where the top cladding is removed, and where a DNA-PAINT target sample can be placed at the interface of the waveguide core and immersed in an aqueous solution containing the complementary imager strands. In the imaging wells, the sample will be illuminated by the waveguide TIRF evanescent field. Our waveguides provide optical sectioning of about 100 nm (Supplementary Fig. 1, Supplementary Fig. 3 and Methods), therefore strongly suppressing the background signal from unbound imager strands (Fig. 1a and Supplementary Fig. 5).

To demonstrate the capabilities of waveguide-PAINT, we imaged microtubules in COS-7 cells as well as DNA origami (Methods). Our waveguides were wide enough to accommodate multiple cells (~100 μm), and in total as many as four whole cells could fit within a single field-of-view. We imaged cells with

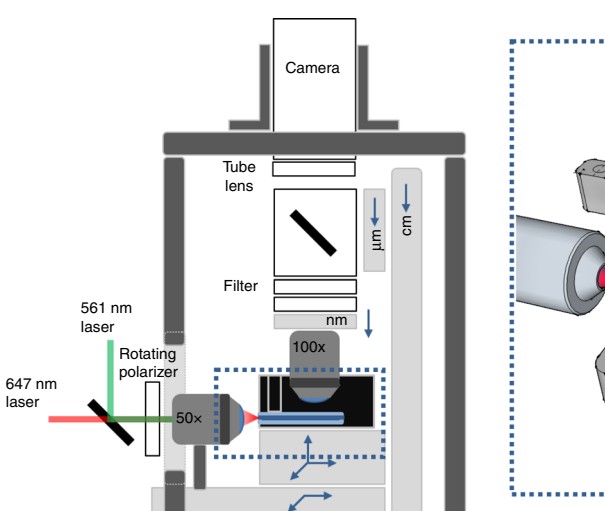
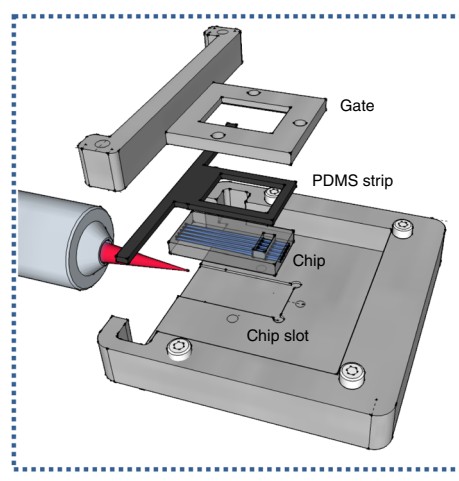

**Fig. 2** Waveguide chip holder and upright microscope for waveguide-PAINT. The proposed DNA-PAINT microscope is designed with two independent arbors: one axle (dark-gray) holds the heaviest components that do not need Z adjustments while a three-gauges system (light gray) provides large (centimeter range), micrometric and nanometric positioning. The waveguide holder (dashed blue line) enables a free space coupling through a *X-Y-Z* nanometric stage placed on top of the *X-Y* stage for FOV adjustments. The holder design presents a precision slot to position and orient the waveguide chip properly with respect to the laser line, and a sealing gate to hold the imaging buffer and to shield scattered light

waveguide-PAINT that were pre-extracted then antibody-stained against tubulin, and observed a continuous network of micro-tubules within each cell, only depleted below the cell nucleus (Fig. 3a, Supplementary Figs. 9, 10 and 11). The structure of individual microtubules can be approximated as a hollow cylinder, of which the antibody labels the outer surface. Thus, the two-dimensional projection of microtubules results in two peaks, separated by the microtubule diameter expanded by the size of the labels. Consistent with this, we measured diameters of ~40 nm (Fig. 3b). We also used this data to quantify the uniformity of the waveguide TIRF excitation by measuring the localization precision across the field of view and performing superresolution optical fluctuation imaging SOFI[50] (Supplementary Fig 12 and 13).

We next used DNA-origami structures to better quantify the performance of waveguide-PAINT. For demonstration purposes, we chose a rectangular origami structure with target strands in a 4 × 3 grid, with a regular spacing of 20 nm. In principle, if their deposition was controlled, a single field of view could fit up to ~$10^5-10^6$ individual structures. In practice, origami were randomly deposited and sometimes sticking together, limiting their surface concentration. In one example field-of-view, we imaged ~1000 origami, which can be readily identified by the 20 nm regular spacing between clusters of localizations (Fig. 3c). Most structures were incomplete, which is consistent with reported folding efficiencies[28]. However, by aligning and over-laying PAINT reconstructions of multiple origami, we recovered both the expected 4 × 3 structure (Fig. 3d) as well as the grid spacing (Fig. 3e and Supplementary Fig. 14).

## Discussion

Increasing the throughput of localization microscopy has pre-viously been achieved by automating acquisitions to image mul-tiple FOV[51] or multiple wells under different conditions[52]. Alternatively, using uniform illumination to image larger FOV can improve both data throughput and quality, since non-uniform illumination results in spatially varying photophysical properties and signal-to-noise ratios[29–31,52]. Creating large, high-quality localization microscopy datasets is essential for leveraging

strategies to determine the molecular organization of structures via single particle averaging[53,54] or particle reconstruction[55]. Yet, little progress has been made toward extending these particular improvements in throughput to PAINT. This is in part because of the optical sectioning requirement of PAINT, and the challenges of achieving large, uniform TIRF illumination. Due to its robustness, the waveguide-PAINT approach we demonstrate here can readily be integrated into an automated system for imaging multiple FOV or multiple waveguides. Datasets acquired by waveguide-PAINT can then be used directly in a particle aver-aging or reconstruction workflow.

We also expect that the precise tunability of waveguide prop-erties, by choice of material and geometry, will give rise to exciting new applications both in PAINT and TIRF imaging in general. For example, by changing waveguide core thickness or index of refraction, one can tune the penetration depth of the evanescent field. Together with simulations to predict waveguide performance, this allows access to a much wider range of illu-mination parameters with waveguide TIRF than is possible with objective TIRF, and in a much more reproducible manner. For example, the surface area generating TIRF illumination can be adapted to the desired geometry, and the waveguide can be designed to produce either multimode or single mode propaga-tion (multimode can be preferable for performing super-resolution optical fluctuation imaging (SOFI)[33].

Although fluorescence imaging has many advantages, even more can be learned by combining measurement modalities[56]. An advantage of our waveguide design for correlative imaging is the use of imaging wells which provide a natural reference frame for aligning different measurements – e.g. for correlative light and scanning electron microscopy[57]. The low surface roughness of the etched wells also make them suitable, in principle, for atomic force microscopy; thus we expect that in the future waveguides will prove useful for correlative PAINT-AFM measurements.

## Methods

**Sample preparation**. COS-7 cells were maintained in DMEM (Life Technologies) supplemented with 10% fetal calf serum (HyClone) and passaged every 3 days. The waveguide chips were cleaned with Hellmanex III (Sigma, 2% in water) at 50 °C for 10', rinsed in water and UV sterilized for 15 min. For imaging, the cells were seeded

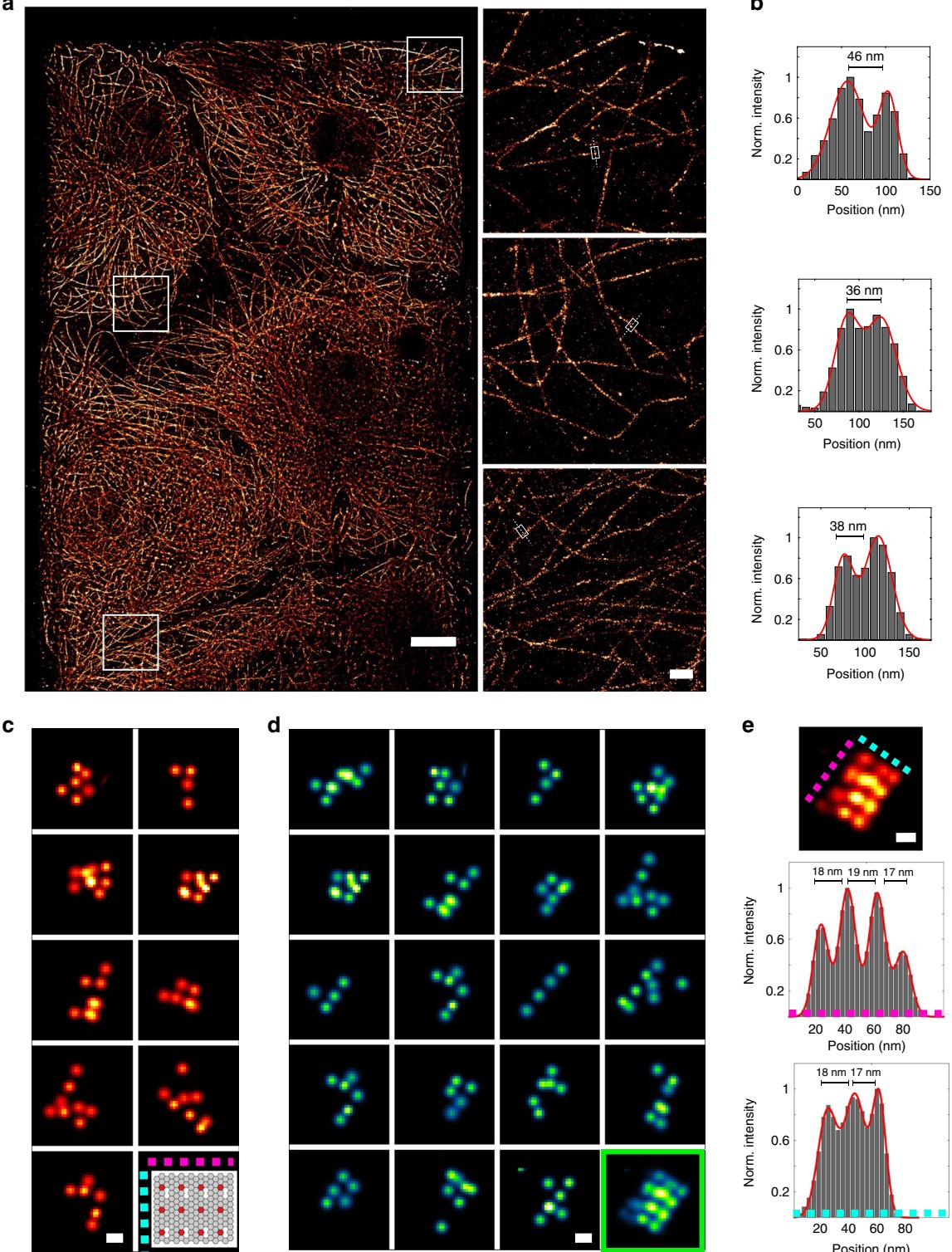

**Fig. 3** Demonstration of waveguide-PAINT. **a** Single FOV DNA-PAINT reconstruction of COS-7 cells cultured on a waveguide, labeled with antibodies against α-tubulin and imaged using a 500 pM concentration of imager strand (I1-655, Ultivue Duplex Kit) (left panel). Magnified views of the boxed regions (left panel, from top to bottom) show microtubules well-resolved across the FOV (**a** right panel). **b** Intensity profiles across individual microtubules (as defined in **a** right panel) reveal two peaks that can be described by the sum of two Gaussian functions (**b** red line). **c** Magnified regions from a single FOV of DNA-PAINT reconstructions of a 20-nm-grid DNA origami imaged with Cy5-conjugated imager strands (500 pM). **d** Aligning single DNA origamis to create an average (green box) reveals the 4 × 3 grid arrangement as well as the respective grid spacing. **e** Intensity profiles along colored axes in **d** indicated in top averaged image. Scale bars: 10 μm (**a**), 0.5 μm (**a** right panel) and 20 nm (**c** and **d**)

on cleaned waveguide chips incubated before with 1% poly-L-lysine (vol/vol) (Sigma) in water for 15 min. After 24 h, the cells were washed in pre-warmed PBS, pre-extracted with 0.5% (vol/vol) Triton X100 (Sigma) in BRB80 buffer (80 mM Pipes, 1 mM MgCl2, 1 mM EGTA) for 15 sec, then fixed with ice-cold methanol for 10 min. The cells were washed in PBS and blocked for 30 min using 5% bovine serum albumin in PBS. Microtubules were labelled for 2 h at room temperature with primary antibodies against α-tubulin (B512, monoclonal produced on mouse, Sigma T6074) diluted 1:100 in PBS supplemented with 0.2% (vol/vol) Triton X100 (PBST). Unbound antibodies were removed in three washing steps with PBST for 10 min each. The samples were incubated with secondary antibodies PBST for 1 h followed by three washes in PBST. We used a combination of DNA-labelled (goat-anti mouse I1, Ultivue Duplex Kit (discontinued), 1:100) and Alexa555-labelled (goat-anti mouse, Life Technologies (A21422), 1:5000) secondary antibodies to allow focusing and selection of the field of view. Finally, the cells were fixed again in Methanol and stored at 4 °C until further use.

For cell membrane imaging, cells were chemically fixed with 4% paraformaldehyde (Alfa Aesar) in PBS for 10 min and then labelled with A647-conjugated cholera toxin B (Life Technologies).

Custom microtubule-like DNA-Origami (reproduced from[11]) were purchased from Gattaquant, the 20 nm DNA origami grid was kindly provided by Ralf Jungmann. DNA origami imaging samples were prepared as described using buffers A (10 mM Tris-HCl and 100 mM NaCl at pH 8.0), A+ (10 mM Tris-HCl, 100 mM NaCl and 0.05% (vol/vol) Tween 20 at pH 8.0) and B+ (5 mM Tris-HCl, 10 mM MgCl2, 1 mM EDTA and 0.05% (vol/vol) Tween 20 at pH 8.0)[28]. Briefly, cleaned waveguides were incubated with BSA-biotin (1 mg/ml in buffer A) for 2 min, then rinsed in buffer A+ and incubated with streptavidin solution (0.5 mg/ml in buffer A). The waveguides were washed sequentially in buffer A+ and B+ before incubation with DNA origamis diluted in buffer B+ for 2 min. Samples were washed again in buffer B+ and stored at 4 °C until further use.

**DNA-PAINT microscope**. Here we described more in detail a workflow (Supplementary Fig. 15) to realize the microscope as well as the microscope components. A description of the main steps to set up the waveguide chip-holder system for proper coupling is described in the Supplementary Note 2. The workflow presents few main steps such as the chip geometry design, the chip-holder design and testing before proceeding with the last stage—DNA-PAINT experiments—where microscope performance is quantified with single molecule localization statistic (see Supplementary Table 2).

The first step of the workflow pipeline is the chip geometry design in order to fit the desired waveguide taper length and waveguide number with both the objective size and the overall stage range. Then, accounting for the chip geometrical parameters, the mechanicals needs (such screw dimensions and positions) and the desirable shield dimension the chip holder have to be defined. Third, the chip-holder system, must be tested to check whether an easy coupling could be established. Finally the holder has to be tested under the microscope to check that all the parts properly fit in the desired range. The waveguide chips needs to be independently tested in order to both identify the ones with the best performance and to improve the chip layout readability (such has chip name position, waveguide identification number position, and inter-waveguide distance). The microscope is equipped with two laser lines of 647 nm (CUBE, Coherent) and 562 nm (OBIS, Coherent) wavelengths coupled with a free space configuration (50X NA 0.55 Mitutoyo, long working distance objective) at the input waveguide facet. A rotating polarizer placed in front of the coupling objective modulates the interference patter when the pure single mode condition is not perfectly established.

The two vertical structures are realized with a two-Z-stage system which holds only the lightest imaging components (the imaging objective mounted on third Z-piezo positioner, the emission filter) while an independent stable mechanical system holds the camera and the tube-lens. Decoupling the Z-movement from the heavy components ensures stability and quick Z-adjustment.

The two-Z-axis system is realized with one long-travel vertical translation stage (VAP10/M, Thorlabs) for large and rough range adjustment and with a second linear translation stage (LNR25D/M, Thorlabs) to ensure micro-z-positioning. The realized independent camera mechanical holder is built with four posts and four rectangular optical breadboards system.

The illumination kit (WFA1010, Thorlabs), equipped with a removable filter holder (CFH2/M, Thorlabs), can be set at one of the two independent column for the epi-channel inspection.

The holder can be easily screwed into the 2-X-stage system where the first stage (a two-axis linear stage, M-401, Newport) enables the FOV adjustment and the second one (a three-axis stage, MAX311D/M, Thorlabs) provides waveguide coupling light alignment. Emitted light from the sample was collected by the objective lens (Water Dipping CFI PLAN × 100 W/NA 1.1, Nikon Nikon or PLAN N 4 ×/NA 0.10 Olympus), then imaged by a tube lens (fTL = 200 mm, Nikon) onto the sCMOS camera (Prime 95B25MM, Photometrics).

**DNA-PAINT imaging and data analysis**. We used three different DNA imager strands: 1) I1-655 (Ultivue Duplex Kit), 2) P3-Atto655: GTAATGAAGA-Atto655[28] and 3) Atto655-imager strand (Gattaquant). PAINT imager strands were

diluted in buffer B+ to a final concentration between 0.1–0.5 nM. Prepared waveguides were inserted into the custom holder and the diluted imager (~2 ml) was added. The holder was fixed on the microscope stage and the position adjusted until the laser coupling was satisfying. The microscope and all components were controlled with μManager[58]. The 642 nm laser output power was set to 5 mW and the sample was moved into focus using the coarse and fine focusing screws. A single waveguide well was centered on the camera and the exposure time was adjusted to maximize signal to noise without creating overlapping localizations (typically around 150 ms). For microtubules (Fig. 3), we acquired 25,000 frames at 150 ms continuous exposure and 100 mW laser output power. DNA origami structures in Fig. 3 were imaged for 10,000 frames at 300 ms.

Single-molecules were localized using ThunderStorm[59] or a recent GPU-based fitting algorithm[60]. Localizations were drift corrected using redundant cross correlation[61], filtered and visualized using a Gaussian-blurred (1xsigma) 2D histogram.

**Chip fabrication**. We fabricated the $Si_3N_4$ waveguide chips in the EPFL Center of Micro-Nanotechnology (CMi). A 2 μm $SiO_2$ cladding layer (refractive index $n_0 \sim$ 1.47 at $\lambda = 647$ nm) was grown by thermal oxidation of the standard silicon (Si) substrate, 525 micrometers thick and 100 mm in diameter. The core layer of 150 nm thick high-stress $Si_3N_4$ ($n_1 \sim 2.04$ at $\lambda = 647$ nm) was then deposited using low-pressure chemical vapor deposition (LPCVD). Electron-beam lithography, followed by dry reactive ion etching (RIE), was then used to define the geometry of the waveguides. A final layer of 2 μm silicon dioxide was deposited by LPCVD (LTO, Low Temperature Oxide) to serve as a protective layer against dust and to reduce unwanted scattering along the waveguide. Another photolithography and dry etching were then used to define an imaging well, where the fluorescent sample should be placed in order to be excited by evanescent waves. To define the waveguide facets as well as the chip borders two more steps of photolithography and etching were needed to spatially disjoin the entrance of the waveguide from the rough chip border. During the first step, smooth input interface was obtained by slow reactive ion etching of $SiO_2/Si_3N_4$ layers on top of Si substrate. In the second step a small margin of 3–4 micrometers was added to the newly formed chip borders and silicon substrate etching was performed using the Bosch process to produce the trenches 300 μm deep. The wafer was then grinded from the back till the chips were split apart using the DAG810 automatic surface grinder. The step-by-step fabrication process is further detailed in Supplementary Table 1 and Supplementary Figures 2, 16, 17 and 18).

**Numerical simulations**. To converge on the design parameters for the waveguide chip layout, we performed multiple numerical simulations using Matlab (in the slab waveguide approximation) and Lumerical FDTD software, which is widely used for the simulation of optoelectronic devices and photonic integrated circuits. To optimize the simulation time the waveguide was divided into three main components: inverted taper for efficient coupling, adiabatic mode expansion taper and a straight waveguide section. As the complexity of simulations grows significantly with the size of simulated area, these three components were analyzed separately.

The simulations of the inverted taper considered only the width of the tip as the variable parameter and the optimal dimensions of the inverted taper tip were found to be around 150 nm at 647 nm illumination wavelength. Two parameters were analyzed in the simulations of the mode expansion taper: its shape (linear, parabolic or quadratic) and expansion rate (change in width divided by length of the taper). Though the linear tapers are most widely used and are the easiest to implement, the non-linear ones allow to reduce the total length of the expansion taper, thus saving wafer space and reducing the cost per waveguide chip. The lowest losses and the most uniform illumination intensity were obtained for the linear taper with an expansion rate <1%.

The extracted parameters were in good agreement with literature and served as the basis for the waveguide chip design, which is described below. The simulation details can be found in the Supplementary Note 1.

**Penetration depth simulation and measurements**. The penetration depth was measured with beads with a diameter of 6.58 μm labelled with Atto647 or with Atto565. The beads were directly dried on top of the waveguide clean surface. The penetration depth was estimated by measuring the individual bead intensity profiles. The measurement is further elucidated in Supplementary Fig. 4.

**Reporting summary**. Further information on experimental design is available in the Nature Research Reporting Summary linked to this article.

## Code availability

Software is available on https://github.com/LEB-EPFL/WGmode.

## Data availability

All data and software used to support the results of this manuscript are available from the Lead Contact upon reasonable request. Original data is available on https://doi.org/10.5281/zenodo.1475055.

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

## Acknowledgements

We would like to thank Alexander Auer and Ralf Jungmann (MPI and LMU) for kindly providing DNA origami samples and discussing experimental conditions, Jürgen Schmied (Gattaquant) for developing a custom DNA origami sample and Christoph Sikeler (EPFL) for experimental support of sample preparation and troubleshooting of PAINT imaging. We also thank Tobias Kippenberg, Ryan Schilling, Martin Pfeiffer (K-Lab, EPFL), and Ligentec SA, for their guidance regarding chip fabrication and Robin Diekmann and Thomas R. Huser (U. Bielefeld) for helpful discussions concerning the microscope setup and waveguide coupling. Chips were fabricated at the EPFL Center of Micronanotechnology (CMi). We acknowledge support from the Swiss National Science Foundation through the National Centre of Competence in Research Bio-Inspired Materials (E.G.) and the National Centre of Competence in Research Chemical Biology (C.S.).

## Author contributions

S.M. conceived the main idea. S.M. and A.R. managed the project. A.A., E.G. and A.S. designed, fabricated and tested the waveguide chips with guidance from A.R. and S.M. A.A. designed the waveguide holder and wrote simulation code. A.A. and CS designed and built the microscope. C.S. and A.A. performed the cell and DNA origami preparation and imaging. S.M. and A.A., with contributions from all authors, wrote the manuscript.

## Additional information

**Competing interests:** The authors declare no competing interests.

