## [Peer Review File · Nature Communications]

Reviewers' comments:

Reviewer #1 (Remarks to the Author):

I have carefully evaluated the manuscript entitled "An open platform for large field-of-view waveguide-PAINT" by Archetti et al. Overall, the manuscript is well written, the analyses are well executed, all procedures are described in sufficient detail and the manuscript is worthy of publication in Nature Communications. I particularly laud the authors for providing open access to some of their detailed drawings and procedures. The paper expands upon some recently published reports on waveguide TIRF excitation for single molecule localization microscopy. There are, however, sufficiently new results reported in this paper, e.g. details on the design and fabrication of the waveguide devices, the addition of a tapered structure for improved coupling and more even illumination of the waveguide structure, and the application to DNA-PAINT, which, in sum, justify publication in Nature Communications.

My only minor comments on this paper are:

1. On page 2, where various implementations of TIRF illumination are reported to justify the need for waveguide excitation, then very successful implementation of ringTIRF illumination by the group of Derek Toomre at Yale University should also be included.

2. Figure 1: what happens at the interface between the top cladding and the imaging well. The abrupt change of the top cladding would likely lead to strong scattering at this interface, doesn't it? A few sentences should be added to discuss the situation at this interface and how, if it is significant, scattering at this interface is excluded or suppressed.

3. In the final sentence of the Discussions section, the relevance of this particular waveguide chip to correlative microscopy, e.g. in combination with AFM is highlighted. An even more important form of correlative microscopy is, however, the combination of super-resolution optical microscopy with forms of electron microscopy, e.g. HR-TEM or cryo-EM, so this should also be mentioned, here.

Reviewer #2 (Remarks to the Author):

In the MS "Waveguide-PAINT: An open platform for large field-of-view super-resolution imaging" by Archetti et al. the authors present a new platform for waveguide based TIRF super-resolution imaging.

One effect of using the waveguide approach presented here is a fairly uniform illumination over the field of view which has until recently not been easy to achieve.

1) In this context, a reader may wonder, why one may have to go to the considerable effort of a waveguide based system if simpler alternatives, e.g. a refractive beam-shaping element, could achieve the same effect. Indeed, the authors make a few good points about this, but tucked away in the last paragraph of the discussion. I wonder if some of these points could not be raised earlier, e.g. in the introduction or similar.

2) The MS very briefly mentions, in a couple of sentences, the recent publication by Diekmann et al. 2017. To this reviewer it seems that there are considerable parallels in the approaches, which is not an issue per se. It would be useful if the differences and distinctions of the approaches presented here, as compared to the Diekmann et al setup, could be emphasised. In particular, I was hoping for differences beyond DNA-PAINT vs dSTORM, which appear to be a minor application aspect - unless the authors can argue otherwise. One aspect that seemed to me addressed in this

MS, as compared to Diekmann et al, is the uniformity of the illumination pattern, e.g. Fig. 1d.

3) The MS makes reference to a dipping objective in the imaging path of the setup but I was unable to find any details about this objective, its mag (maybe 100x from Fig. 2) and NA in particular. Did I miss something in the methods?

4) How was the localisation precision calculated in sup fig 12? If using some software please provide the formula underlying it. Also, it should be noted that the simple formulae provide a best case scenario for the actual localisation error which is in practice often larger (deviations from ideal models underlying formulae). If providing the localisation precision the reliability of these estimates should be tested with the origami data where the loc res can be determined from the std dev of localisations around individual 'dots' on the origami.

5) The discussion says "this allows access to a much wider range of illumination parameters with waveguide-TIRF than is possible with objective- TIRF, and in a much more reproducible manner.". It was not obvious to this reviewer what benefits this "much wider range of illumination parameters" might entail in practice. Some concrete examples would be helpful.

6) The MS provides a link to the data which was not accessible (10.5281/zenodo.1475055), it would be useful if this was accessible to reviewers.

Response to Reviewers

Waveguide-PAINT: An open platform for large field-of-view super-resolution imaging

Reviewer #1 (Remarks to the Author)

I have carefully evaluated the manuscript entitled "An open platform for large field-of-view waveguide-PAINT" by Archetti et al. Overall, the manuscript is well written, the analyses are well executed, all procedures are described in sufficient detail and the manuscript is worthy of publication in Nature Communications. I particularly laud the authors for providing open access to some of their detailed drawings and procedures. The paper expands upon some recently published reports on waveguide TIRF excitation for single molecule localization microscopy. There are, however, sufficiently new results reported in this paper, e.g. details on the design and fabrication of the waveguide devices, the addition of a tapered structure for improved coupling and more even illumination of the waveguide structure, and the application to DNA-PAINT, which, in sum, justify publication in Nature Communications.

We thank the reviewer for their careful reading of our manuscript, and for their positive evaluation of the results we describe. We especially appreciate that the reviewer examined the waveguide design and fabrication details, finding their open access relevant. We have addressed the reviewer's remaining concerns with the following changes to the manuscript:

- 1. On page 2, where various implementations of TIRF illumination are reported to justify the need for waveguide excitation, then very successful implementation of ringTIRF illumination by the group of Derek Toomre at Yale University should also be included.*

We agree, and ringTIRF illumination is now referenced in the introduction (refs 5 and 6 relate to ringTIRF):

"Optical sectioning can be provided by confocal rejection¹ or total internal reflection fluorescence (TIRF)². However, confocal rejection also reduces the number of detected signal photons, while TIRF is typically limited in both size and uniformity of illumination. Note that even sophisticated TIRF setups using scanning of the coherent excitation light to reduce interference patterns³⁻⁶ do not eliminate the spatial dependence of the field resulting from a focused Gaussian beam or the field-of-view (FOV) limitation".

- 2. Figure 1: what happens at the interface between the top cladding and the imaging well. The abrupt change of the top cladding would likely lead to strong scattering at this interface, doesn't it? A few sentences should be added to discuss the situation at this interface and how, if it is significant, scattering at this interface is excluded or suppressed.*

The reviewer is right about this point, although the top cladding is etched away to form the imaging well before the sample is introduced. Thus, the scattering is strongly localized

at the border, and can be easily excluded by limiting the ROI to an area that excludes the well edges - few microns smaller than the well itself, less than 1% of the total imaging area. This cropping of the data barely reduces the size of the FOV.

To show the steep decay of intensity at the well edge, we added Supplementary Figure 17.

- 3. In the final sentence of the Discussions section, the relevance of this particular waveguide chip to correlative microscopy, e.g. in combination with AFM is highlighted. An even more important form of correlative microscopy is, however, the combination of super-resolution optical microscopy with forms of electron microscopy, e.g. HR-TEM or cryo-EM, so this should also be mentioned, here.*

We thank the reviewer for this suggestion. For the implementation of correlative SEM and AFM imaging, the waveguide fabrication process would require no changes. However, for TEM imaging the sample support would need to be made transparent to electrons, similarly as in silicon nitride support films (also called Si₃N₄ TEM membranes). We now added one sentence (in bold) at the end of the Discussion paragraph:

“An advantage of our waveguide design for correlative imaging is the use of imaging wells which provide a natural reference frame for aligning different measurements – e.g for **correlative light and scanning electron microscopy**⁷. The low surface roughness of the etched wells also make them suitable, in principle, for atomic force microscopy; thus we expect that in the future waveguides will prove useful for correlative PAINT-AFM measurements.”

We would like to thank the reviewer once more for critically examining our manuscript and providing feedback that has undoubtedly improved its quality.

Reviewer #2 (Remarks to the Author)

In the MS "Waveguide-PAINT: An open platform for large field-of-view super-resolution imaging" by Archetti et al. the authors present a new platform for waveguide based TIRF super-resolution imaging.

One effect of using the waveguide approach presented here is a fairly uniform illumination over the field of view which has until recently not been easy to achieve.

We thank the reviewer for their comments and for their positive assessment of our approach. We agree that there were some important details missing from the discussion of our approach, and we have addressed the reviewer's concerns as follows:

- 1. In this context, a reader may wonder, why one may have to go to the considerable effort of a waveguide based system if simpler alternatives, e.g. a refractive beam-shaping element, could achieve the same effect. Indeed, the authors make a few good points about this, but tucked away in the last paragraph of the discussion. I wonder if some of these points could not be raised earlier, e.g. in the introduction or similar.*

We thank the reviewer for this suggestion. We now added in the introduction this sentence:

"The waveguide TIRF approach, compared with other approaches such as refractive beam-shaping elements⁸⁻¹⁰, introduces additional flexibility including the freedom to image with a low magnification objective¹¹ (Fig. 1a) and the generation of an evanescent field with a uniform penetration depth^{12,13}, as well as built-in reference markings for correlative measurements."

- 2. The MS very briefly mentions, in a couple of sentences, the recent publication by Diekmann et al. 2017. To this reviewer it seems that there are considerable parallels in the approaches, which is not an issue per se. It would be useful if the differences and distinctions of the approaches presented here, as compared to the Diekmann et al setup, could be emphasised.*

We agree with the reviewer, and compiled the following list of differences:

- Our chip is optimized for single mode excitation
- Our microscope addresses 3-axis manipulation for straightforward alignment in a compact and customizable solution, whereas Diekmann's is built around a commercial microscope
- Our chip holder makes it easy to couple the waveguide due to an engineered slot for positioning and to keep the chip alignment with the input beam for consecutive experiments, as well as a gate to suppress the strong scatter light generated at the coupling input facet. Diekmann's holder uses a vacuum (Waveguide Mount HWV001 Thorlabs), more prone to vibration and without built-in alignment.
- Importantly for PAINT, our chip holder allows for easy buffer exchange.
- We share the microscope, chip and chip holder designs that will allow others to access our approach.

Since many of these points were included in the previous draft, and we did not wish to minimize the findings of Diekmann, we have not expanded on those contrasts. We now added one sentence at the beginning of the Results, Design and construction of a waveguide holder and microscope for PAINT, paragraph:

“To enable waveguides to be used for PAINT, we needed to build a chip holder that would allow access to the entrance window of the waveguides for coupling with excitation light, but also hold the liquid imaging buffer over the sample to allow fluorophore exchange. Ideally, the holder would also facilitate alignment for easy coupling, and prevent scattered light from entering the detection path. **In the previously proposed waveguide platform¹² the chip was secured with a vacuum holder (Waveguide Mount HWV001 Thorlabs) which is more prone to vibration and devoid of the aforementioned features.**”

In particular, I was hoping for differences beyond DNA-PAINT vs dSTORM, which appear to be a minor application aspect - unless the authors can argue otherwise. One aspect that seemed to me addressed in this MS, as compared to Diekmann et al, is the uniformity of the illumination pattern, e.g. Fig. 1d.

The reviewer is exactly right about the comparison with Diekmann work: the main effect on the final image quality is the achievement of the uniformity of the illumination pattern. However, our work also provides the other improvements listed above. As one might expect, we started out by trying to reproduce the results described in Diekmann, and only after the significant improvements to design and fabrication described in our manuscript were we able to arrive at the described results.

- 3. The MS makes reference to a dipping objective in the imaging path of the setup but I was unable to find any details about this objective, its mag (maybe 100x from Fig. 2) and NA in particular. Did I miss something in the methods?*

We thank the reviewer for noticing this missing information. We added now a sentence to read:

“Emitted light from the sample was collected by the objective lens (Water Dipping CFI PLAN $\times 100W/NA$ 1.1, Nikon Nikon or PLAN N 4x/NA 0.10 Olympus), then imaged by a tube lens (fTL = 200 mm, Nikon) onto the sCMOS camera (Prime 95B25MM, Photometrics).”

- 4. How was the localisation precision calculated in sup fig 12? If using some software please provide the formula underlying it.*

We agree that this information was missing.

Also, it should be noted that the simple formulae provide a best case scenario for the actual localisation error which is in practice often larger (deviations from ideal models underlying formulae). If providing the localisation precision the reliability of these estimates should be tested with the origami data where the loc res can be determined from the std dev of localisations around individual 'dots' on the origami.

We thank the reviewer for the suggestion of comparing the estimated localization precision with the std dev of localizations around individual “dots” on the origami.

We added now the Supplementary Figure 18 to address this point.

5. *The discussion says “this allows access to a much wider range of illumination parameters with waveguide-TIRF than is possible with objective-TIRF, and in a much more reproducible manner.”. It was not obvious to this reviewer what benefits this “much wider range of illumination parameters” might entail in practice. Some concrete examples would be helpful.*

We now modified the sentence in the Discussion paragraph:

“Together with simulations to predict waveguide performance, this allows access to a much wider range of illumination parameters with waveguide-TIRF than is possible with objective-TIRF, and in a much more reproducible manner. **For example, the surface area generating a TIRF illumination can be adapted to the desired geometry, the wells design can be modified to suit for different screening conditions and the waveguide mode properties can be designed to produce either a multimode or single mode condition depending on the need (multimode can be suitable to perform super-resolution optical fluctuation imaging (SOFI)¹²).**”

6. *The MS provides a link to the data which was not accessible (10.5281/zenodo.1475055), it would be useful if this was accessible to reviewers.*

We thanks the reviewer for asking to make the link accessible. The data are now available.

We would like to thank the reviewer once more for critically examining our manuscript and providing feedback that has undoubtedly improved its quality.

CHANGES TO THE MAIN MANUSCRIPT

We agreed with all of the reviewer critiques and therefore made all the necessary changes and additions to the manuscript to address their comments.

The following changes were made to the main manuscript:

Addresses Reviewer #1's

Page 2:

Old sentence:

“Note that even sophisticated TIRF setups using scanning of the coherent excitation light to reduce interference patterns^{3,14} do not eliminate the spatial dependence of the field resulting from a focused Gaussian beam.”

New sentence:

“Note that even sophisticated TIRF setups using scanning of the coherent excitation light to reduce interference patterns³⁻⁶ do not eliminate the spatial dependence of the field resulting from a focused Gaussian beam or the field-of-view (FOV) limitation.”

Page 12:

Old sentence:

“An advantage of our waveguide design for correlative imaging is the use of imaging wells which provide a natural reference frame for aligning different.”

New sentence:

“An advantage of our waveguide design for correlative imaging is the use of imaging wells which provide a natural reference frame for aligning different measurements – e.g. for correlative light and scanning electron microscopy⁷.”

Addresses Reviewer #2's

Page 3:

Old sentence:

“Unlike objective-based TIRF which couples an off-axis excitation laser through a high numerical-aperture imaging objective, waveguide-based TIRF decouples excitation and detection paths. This allows independent control over the propagation of excitation light, and independent design of the detection optics¹¹ (Fig. 1a). Thus, waveguide-based TIRF offers the possibility of a larger and more uniform evanescent field^{12,13}.”

New sentence:

“The waveguide TIRF approach, compared with other approaches such as refractive beam-shaping elements⁸⁻¹⁰, introduces additional flexibility including the freedom to image with a low magnification objective¹¹ (Fig. 1a) and the generation of an evanescent field with a uniform penetration depth^{12,13}, as well as built-in reference markings for correlative measurements.”

Page 7:

New sentence:

“In the previously proposed waveguide platform¹² the chip was secured with a vacuum holder (Waveguide Mount HWV001 Thorlabs) which is more prone to vibration and devoid of the aforementioned features.”

Page 11:

New sentence:

“For example, the surface area generating a TIRF illumination can be adapted to the desired geometry, the wells design can be modified to suit for different screening conditions and the waveguide mode properties can be designed to produce either a multimode or single mode condition depending on the need (multimode can be suitable to perform super-resolution optical fluctuation imaging (SOFI)¹²).”

Page 15:

New sentence:

“Emitted light from the sample was collected by the objective lens (Water Dipping CFI PLAN $\times 100W/NA$ 1.1, Nikon Nikon or PLAN N 4x/NA 0.10 Olympus), then imaged by a tube lens (fTL = 200 mm, Nikon) onto the sCMOS camera (Prime 95B25MM, Photometrics).”

Changes to figure:

We noticed an error in the scale bar of Fig. 1f. We updated the caption.

Page 6:

Old caption:

Figure 1 Scale bars: 1 μm (c top and bottom), 10 μm (d left and right) and 500 μm (f).

New caption:

Figure 1 Scale bars: 1 μm (c top and bottom), 10 μm (d left and right) and 200 μm (f).

CHANGES TO THE SUPPLEMENT

The following changes were made to the Supplementary Information

Addresses Reviewer #1's

We now added Supplementary Fig. 17

Addresses Reviewer #2's

We now added Supplementary Fig. 18

We now modify the caption of Supplementary Fig. 12

Old Caption:

“Supplementary Figure 12 Localization precision across the waveguide width. a, Superresolved image section across the waveguide. b, Corresponding scatter-plot of the localizations with color-map proportional to the localization precision. Scale bar 10um.”

New Caption:

“Supplementary Figure 12 Localization precision across the waveguide width. a, Superresolved image section across the waveguide. b, Corresponding scatter-plot of the localizations with color-map proportional to the localization precision. The localization precision is computed using 3 which represents the best case scenario for the actual localization error. Scale bar 10um.”

REFERENCES

1. Schueder, F. *et al.* Multiplexed 3D super-resolution imaging of whole cells using spinning disk confocal microscopy and DNA-PAINT. *Nat. Commun.* **8**, (2017).
2. Jungmann, R. *et al.* Single-molecule kinetics and super-resolution microscopy by fluorescence imaging of transient binding on DNA origami. *Nano Lett.* **10**, 4756–4761 (2010).
3. Mattheyses, A. L., Simon, S. M. & Rappoport, J. Z. Imaging with total internal reflection fluorescence microscopy for the cell biologist. *J. Cell Sci.* **123**, 3621–3628 (2010).
4. Mattheyses, A. L. & Axelrod, D. Direct measurement of the evanescent field profile produced by objective-based total internal reflection fluorescence. *J. Biomed. Opt.* **11**, 014006 (2006).
5. Yang, Q., Karpikov, A., Toomre, D. & Duncan, J. S. 3-D reconstruction of microtubules from multi-angle total internal reflection fluorescence microscopy using bayesian framework. *IEEE Trans. Image Process.* **20**, 2248–2259 (2011).
6. Roorda, R. & Toomre, D. AN OPTICAL SYSTEM FOR ILLUMINATION OF AN EVANESCENTFIELD. *Patent* 050743 A2 (2007).
7. Kopek, B. G., Shtengel, G., Grimm, J. B., Clayton, D. A. & Hess, H. F. Correlative Photoactivated Localization and Scanning Electron Microscopy. *PLoS One* **8**, (2013).
8. Rowlands, C. J., Ströhl, F., Ramirez, P. P. V., Scherer, K. M. & Kaminski, C. F. Flat-Field Super-Resolution Localization Microscopy with a Low-Cost Refractive Beam-Shaping Element. *Sci. Rep.* **8**, 1–8 (2018).
9. Khaw, I. *et al.* Flat-field illumination for quantitative fluorescence imaging. *Opt. Express* **26**, 15276 (2018).
10. Zhao, Z., Xin, B., Li, L. & Huang, Z.-L. High-power homogeneous illumination for super-resolution localization microscopy with large field-of-view. *Opt. Express* **25**, 13382 (2017).
11. Agnarsson, B., Ingthorsson, S., Gudjonsson, T. & Leosson, K. Evanescent-wave fluorescence microscopy using symmetric planar waveguides. *Opt. Express* **17**, 5075 (2009).
12. Diekmann, R. *et al.* Chip-based wide field-of-view nanoscopy. *Nat. Photonics* 1–9 (2017). doi:10.1038/nphoton.2017.55
13. Tinguely, J.-C., Helle, Ø. I. & Ahluwalia, B. S. Silicon nitride waveguide platform for fluorescence microscopy of living cells. *Opt. Express* **25**, 27678 (2017).
14. Mattheyses, A. L., Shaw, K. & Axelrod, D. Effective Elimination of Laser Interference Fringing in

Fluorescence Microscopy by Spinning Azimuthal Incidence Angle. *Microsc. Res. Tech.* **69**, 642–647 (2006).

REVIEWERS' COMMENTS:

Reviewer #2 (Remarks to the Author):

The authors have responded to all my comments satisfactorily. I support publication of this revised version.